# Decoding Ribosome Heterogeneity: A New Horizon in Cancer Therapy

**DOI:** 10.3390/biomedicines12010155

**Published:** 2024-01-11

**Authors:** Valerio Gelfo, Giulia Venturi, Federico Zacchini, Lorenzo Montanaro

**Affiliations:** 1Department of Medical and Surgical Sciences (DIMEC), University of Bologna, 40138 Bologna, Italy; valerio.gelfo2@unibo.it (V.G.); giulia.venturi13@unibo.it (G.V.); 2Centre for Applied Biomedical Research (CRBA), Bologna University Hospital Authority St. Orsola-Malpighi Polyclinic, 40138 Bologna, Italy; 3IRCCS Azienda Ospedaliero-Universitaria di Bologna, 40138 Bologna, Italy; federico.zacchini@aosp.bo.it

**Keywords:** ribosome, heterogeneity, cancer, target therapy, translation, regulation

## Abstract

The traditional perception of ribosomes as uniform molecular machines has been revolutionized by recent discoveries, revealing a complex landscape of ribosomal heterogeneity. Opposing the conventional belief in interchangeable ribosomal entities, emerging studies underscore the existence of specialized ribosomes, each possessing unique compositions and functions. Factors such as cellular and tissue specificity, developmental and physiological states, and external stimuli, including circadian rhythms, significantly influence ribosome compositions. For instance, muscle cells and neurons are characterized by distinct ribosomal protein sets and dynamic behaviors, respectively. Furthermore, alternative forms of ribosomal RNA (rRNAs) and their post-transcriptional modifications add another dimension to this heterogeneity. These variations, orchestrated by spatial, temporal, and conditional factors, enable the manifestation of a broad spectrum of specialized ribosomes, each tailored for potentially distinct functions. Such specialization not only impacts mRNA translation and gene expression but also holds significant implications for broader biological contexts, notably in the realm of cancer research. As the understanding of ribosomal diversity deepens, it also paves the way for exploring novel avenues in cellular function and offers a fresh perspective on the molecular intricacies of translation.

## 1. Introduction

Ribosomes play a pivotal role in the flow of genetic information by translating mRNAs into proteins across all living organisms [1]. This conserved function is mirrored in their structural composition, which consists of a ribonucleoprotein complex encompassing ribosomal proteins (RPs) and ribosomal RNAs (rRNAs) [2]. In humans, the ribosome is formed of over 80 distinct RPs and four different rRNAs, forming a small (40S) subunit and a large (60S) subunit [3].

A series of studies over the past few decades suggested a potential correlation between alterations in RPs and/or rRNAs and cancer progression. The term “oncoribosome” has been introduced to highlight the perceived heterogeneity of ribosomes in the context of cancer. Moreover, ribosomes are not merely viewed as protein synthesis machinery but as potential contributors to tumorigenesis [4]. Changes in RPs and rRNAs are not just events that passively occur in cancer cells but instead are active participants in the carcinogenic process [5,6,7]. Indeed, as observed in various cancers, such defects in ribosome biogenesis and function can trigger a spectrum of cellular outcomes, from apoptosis to uncontrolled proliferation [7,8,9].

The molecular mechanisms behind these phenomena are complex. While alterations in RPs and rRNAs can disrupt regular cellular processes, leading to ribosomal stress [5,9,10], they might also provide a proliferative advantage to cancerous cells, allowing them to thrive under adverse conditions. There is an expanding body of evidence on the amplification, mutation, and deletion of RPs, which primarily affects ribosome biogenesis and protein biosynthesis [6,9,11].

An additional layer of complexity is due to RP and rRNA modifications that are essential for the proper functioning of ribosomes. Over 14 types of chemical modifications are known to occur on rRNA, such as methylation, pseudouridylation, and base modifications, affecting over 200 sites [12,13]. These modifications play crucial roles in ribosome assembly, stability, and activity [14]. Specifically, pseudouridylation is significant, accounting for approximately 1.4% of all base modifications with a total of 106 predicted sites in human rRNAs [15,16].

Furthermore, the role of rRNA alteration in cancer is an emerging focus of research. Although mutations in RPs are established carcinogenic agents, the role of rRNA mutations remains relatively uncharted. The conventional perspective of the ribosome as a static unit has evolved such that it is now viewed as a dynamic apparatus responsive to specific cellular conditions, adjusting its protein composition for selective mRNA translation [1,16]. Correspondingly, it is postulated that cancer cells might harbor specialized ribosomes, termed ”oncoribosomes”, to facilitate protein synthesis [17,18,19]. This notion of cancer-specific ribosomal heterogeneity raises the important question of its potential therapeutic exploitation.

The advent of advanced ribosomal characterization techniques holds promise for tailored therapeutic interventions [20]. This research trajectory broadens the implications of ribosomal studies beyond oncology, encompassing other pathologies such as neurodegenerative diseases [9].

In conclusion, our perception of ribosomes demands reconsideration. They are not merely functional cellular units but dynamic entities with roles that can either promote or inhibit carcinogenesis. As research progresses, a nuanced understanding of these roles will not only enhance our comprehension of cancer but also pave the way for novel therapeutic avenues.

## 2. Ribosome Heterogeneity in Normal Cellular Function

The traditional understanding of ribosomes as uniform entities in cellular translation is undergoing a paradigm shift. Emerging research suggests the existence of specialized ribosomes with unique compositions and functions [21,22].

So far, several different potential mechanisms are believed to contribute to ribosome heterogeneity [23]. For example, quantification of 15 core RPs in polysomes from mouse embryonic stem cells revealed that 6 of the 15 RPs measured were substoichiometric, with 4 of those present on only 60–70% of polysomal ribosomes, indicating the existence of actively translating ribosomes lacking at least one core ribosomal protein [24].

Cellular and tissue specificities further accentuate this heterogeneity. For instance, muscle cells might possess a distinct ribosomal protein repertoire compared with other cells, thereby modulating ribosomal functions [17]. Also, in neurons, the dynamic behavior of ribosomal proteins showed that they can be rapidly and selectively incorporated into existing ribosomes. This dynamic exchange is context-dependent and varies based on the subcellular location and physiological conditions. In detail, it has been observed that a group of RPs was associated with rapidly translating ribosomes in the cytoplasm, and the incorporation probability of some RPs was regulated by their location (neurites vs. cell bodies) and changes in the cellular environment, such as in response to oxidative stress [25].

The developmental and physiological states of the cell further contribute to this complexity. As cells undergo differentiation, growth, or environmental adaptations, ribosomal compositions may evolve dynamically [26]. RNA-seq analyses across mouse and human tissues and cell lines have indicated a broader variety in RP expression patterns than previously anticipated [27,28]. A case in point is the ribosomal protein RPL38, which is implicated in the specialized translation of *Hox* genes, vital for mammalian development. It does this by aiding the formation of the 80S complex on these specific mRNA molecules, thereby affecting transcript-specific translation. Studies using 80S cryo-electron microscopy have pinpointed the location of RPL38 on the ribosomal surface, close to an area of rRNAs known as expansion segment 27. This particular ribosomal area is highly dynamic and undergoes various shape changes that seem to be regulated by RPL38. This evidence implies that RPL38 may influence the translation specificity of certain mRNA subsets by controlling unique structural alterations in the ribosome [27].

Furthermore, evidence from zebrafish embryos indicates variations in rRNA types during embryogenesis, namely, 5.8S, 18S, and 28S rRNA, and in silico analyses indicate that the 5′UTR of the maternal transcript might preferentially bind to certain regions of the expansion segment that are present in the maternal 18S rRNA subtype but not in the somatic type [29]. Also, in mice and human cells that express reduced levels of the rRNA pseudouridine synthase diskerin, the translation of p27 and p53 is impaired [30].

External stimuli, including physiological cues such as circadian rhythms, can also shape ribosome composition. For instance, Sinturel et al. suggested a coordination between ribosome biogenesis and daily rhythms, suggesting a circadian influence on ribosomal functions in metabolically active tissues [31]. Additionally, rRNA sequence variants and their posttranscriptional modifications introduce another dimension to ribosomal heterogeneity, which is believed to be systematically influenced by spatial, temporal, and conditional determinants [19,21,32], allowing for the existence of a wide array of specialized ribosomes, each with potentially unique functions. Improved rDNA locus mapping has suggested extensive human and mouse RNA sequence variation both across and within individuals. Also, certain rRNA alleles display tissue-specific expression in mice [33]. The functional consequences of the rRNA variations have not yet been established; one hypothesis is that they may contribute to the preferential recruitment of specific mRNAs [29], suggesting that rRNAs may have a direct role in the regulation of translation. Ribosomal heterogeneity does not only occur at the cellular and tissue levels. A single cell may simultaneously contain various subpopulations of ribosomes that differ in protein and rRNA composition and modification. This can alter the structure and function of the ribosome, leading to functional variety within the cell. Understanding these factors opens new avenues for exploring how ribosome specialization can modulate cellular functions, including the control of mRNA translation and gene expression. The implications of specialized ribosomes extend far beyond cellular differentiation and tissue development, reaching into the critical area of cancer research. Emerging evidence suggests that specialized ribosomes may play a pivotal role in the selective translation of oncogenes or tumor suppressor genes, thereby influencing tumorigenesis and cancer progression [34,35]. This information could be invaluable for the development of personalized cancer therapies aimed at modulating ribosomal function.

## 3. Oncoribosome

Ribosomepathies are generally defined as diseases caused by mutations in RPs or factors associated with rRNA transcription and processing, resulting in alterations to ribosome production, assembly, or function [9]. In first-place-impaired ribosome production, decreased cell proliferation generates ribosomal stress, leading to apoptosis in ribosome-defective cells [5,36,37]. Many of these disorders are characterized by an increased susceptibility to cancer [38].

Nonetheless, oncogenic signaling and chronic inflammation are generally associated with an increase in ribosome biogenesis [39,40,41], leading to enhanced translation, cancer-oriented translation, proliferation, cell survival, and resistance to cancer therapy [42,43].

The paradoxical transition from hypo-proliferative phenotypes to hyper-proliferative states in cancer is known as Dameshek’s riddle and remains unresolved [44]. One hypothesis is that cells with high proliferative potential accumulate compensatory genetic defects that produce a cellular advantage and become drivers of transformation. Also, defective ribosomes could contribute to the expression of a different set of genes, favoring the production of cancer-promoting proteins [45].

A link between intrinsic ribosomal alterations and cancer progression was established early in the 21st century [46,47]. Nowadays, a substantial body of evidence exists to suggest that both inherited and acquired defects in ribosomes play a role in shaping cancer characteristics [48,49]. These processes include the cell cycle, cell proliferation, neoplastic transformation, migration, and invasion [50]. Both ribosomal RNA and proteins are subject to mutations, expression level variation, and structural alteration [51]. For a comprehensive overview, refer to Table 1.

The *RPL5* gene, crucial to ribosome biogenesis, is linked to various cancers. Mutations in *RPL5* disrupt ribosome formation and activate the p53 protein by inhibiting HDM2 through the RPL5/RPL11/5S rRNA complex. This mechanism serves as a key defense against cancer. Mutations in RPL5 impair the upregulation of normal p53 and hinder ribosome biogenesis, underscoring their significant role in the development of cancers that retain normal p53 function [52]. In total, 139 *RPL5* mutations and 74 different cancer-associated *RPL11* mutations have been detected across 49 cancer types; these genes were found to be mutated in 34% of breast cancers, 28% of melanomas, and up to 34% of multiple myelomas in a comprehensive analysis of 19,000 cancer samples. The majority of these *RPL5* and *RPL11* mutations are missense mutations (*RPL5*, 66%; *RPL11*, 73%) [52].

Somatic mutations in *RPL10*, namely, R98S, R98C, and Q123P, have an active role in tumorigenesis. The R98S mutation enhances the JAK-STAT signaling pathway, which is vital for cell proliferation and survival. This hyper-activation of the JAK-STAT pathway upon cytokine stimulation, along with increased sensitivity to JAK-STAT inhibitors, indicates that the R98S mutation functionally mimics JAK-STAT activation [53]. In this context, the R98S mutation has been described as a mutation hotspot with >90% of *RPL10* mutations accounting for 8% of all pediatric T-ALL [54,55].

The truncating germline mutation in *RPS20* is associated with microsatellite-stable colon cancer. Mechanistically, the mutation is responsible for a defect in pre-ribosomal RNA maturation [56].

Research indicates that the loss of a single allele of *RPL22* can accelerate T-cell lymphomagenesis, while the loss of both alleles affects the migration ability of cancer cells and confines them to the thymus. The loss of one copy of *RPL22* promotes lymphomagenesis and the spread of the disease. This effect is due to the activation of the NF-κB pathway and the induction of the stem cell factor Lin28B. Paradoxically, the loss of both alleles of *RPL22* restricts the progression of lymphoma, particularly affecting the migratory ability of malignant cells out of the thymus. This can be traced back to the downregulation of the *KLF2* transcription factor and its targets, including the key migratory factor sphingosine-1-phosphate receptor 1 (*S1PR1*). Re-expressing *S1PR1* in *RPL22*-deficient tumor cells restores their migratory capacity in vitro [57]. Also, *RPL22* has been found to be mutated in 10% of gastric, endometrial, and colorectal cancers [58,59,60].

The *RPL23A* gene is amplified in 12.5% of uterine cancer cases. In one study, patient samples harboring *RPL23A* amplification displayed 1.5-fold higher average *RPL23A* mRNA expression levels compared with *RPL23A* diploid tumor samples [61].

Ntoufa et al. showed that, in primary chronic lymphocytic leukemia (CLL) cells, mutant RPS15 displayed altered translation efficiency in other ribosomal proteins and regulatory elements, affecting key cell processes such as translational machinery and immune signaling, as well as genes implicated in CLL [62].

**Table 1 biomedicines-12-00155-t001:** Alteration of ribosomal components in cancer.

Ribosomal Component	Type of Change	Ribosomal Protein	Phenotypic Consequences
Ribosomal proteins	Mutation(s)	RPL5 and RPL11	Forty-nine cancer types [61].
		RPL10	T-ALL and breast cancer [54,63].
		RPS20	Colorectal cancer [56].
		RPL22	Endometrial, colorectal, gastric, breast, and non-small-cell lung cancer [64].
		RPL23A	Uterine cancer [61].
		RPS15	Primary chronic lymphocytic leukemia [62].
	Expression Level Variation	Increased/decreased RPS9, RPS14, RPL5, RPL10, RPL11, RPL15, and RPL39	Initiation, progression, promotion of metastasis, regulation of translation, and cell cycles in circulating breast tumor cells [65].
		Increased RPL17	Colorectal cancer [66].
		Increased RPS3	Impacts protein synthesis regulation and cancer progression in colorectal cancer [67].
		Increased RPS6 and RPL19	Poor prognosis in cancer [40].
Ribosomal RNA	Mutation(s)	5S rRNA tip	Impairs the interaction between Rpf2–Rrs1 and 5S rRNA [68].
		18S rRNA (1248.U)	Present in 45.9% of CRC patients and across >22 cancer types [69].
	Post-Translational Modification	2’-*O*-Methylation	- Enhanced translation of IGF-1R, C-MYC, VEGF-A, and FGF1/2 [70,71]. - Tumor development and apoptosis in CRC [72]. - Cancer promotion or suppression in lung cancer and AML [70,73]. - Ribosome heterogeneity: altered translation of mRNAs related to cell cycle, mitosis metabolism, oxidation–reduction, and intracellular transport [74]. - Development stage of acute myeloid leukemia and its gene expression signature [73].
		Pseudouridylation	- Impairment of translational fidelity and increased IRES-dependent translation of p27 tumor suppressor, XIAP, TP53, CDKN1B, and BCL2L1 [30,71,75]. - Lymph node metastasis at diagnosis, vascular infiltration by tumor cells, p53 accumulation, and BCL-2 and EGFR expression [13].
	Expression Level Variation	Increased rRNA levels	- Hypomethylation at the rDNA promoter region accompanied by chromatin decondensation in rDNA in human cervical cancer [76]. - Linked to prostate cancer [77]. - rRNA transcripts dysregulated in breast cancer [78].
	Structural Alteration	Truncation of 23S rRNA	Dysfunctional ribosome [79].
	Environmental Influence	Therapeutic agents	- Doxorubicin induces RNA disruption [80]. - 5-FU is incorporated into rRNAs of mature ribosomes, inducing altered mRNA translation [81].

Expression level variations in ribosomal proteins have a key role in cancer. One illustrative example is represented by an increased expression level of RPL5 in colon cancer, revealing its significant contribution to proliferation and migration. In an in vitro study, Zhang et al. revealed that the expression level of RPL5 was higher in HCT116 cells and RKO cells compared with adjacent tissues and NCM460 cells.

Reducing RPL5 levels hinders the growth and migration of colon cancer cells and causes cell cycle arrest. This effect is linked to changes in the MAPK/ERK signaling pathway, including reduced levels of p-MEK1/2, p-ERK, c-Myc, and increased FOXO3. Additionally, an ERK activator (TBHQ) can partially reverse these effects, suggesting RPL5’s role in promoting colon cancer cell proliferation and migration via this pathway [82]. Also, an increase in RPL10 expression has been observed in ovarian cancers and linked to enhanced cell proliferation, invasion, survival, resistance to oxidative stress, and decreased apoptosis [83]. A comprehensive list of expression level variations in RPs in cancer is reported in Table 1.

These findings strengthen the link between mutations and expression level variation in ribosomal proteins and cancer progression, offering valuable insights into cancer mechanisms and potential therapeutic targets.

In addition, the catalytic core of the ribosome is functionalized by a constellation of at least 14 different chemical modifications across >200 sites [13]. Pseudouridylation and 2′-*O*-methylation are the most represented post-transcriptional modifications in rRNA, primarily mediated by small nucleolar RNAs (snoRNAs). These modifications significantly impact ribosome functionality. For instance, they enhance the structural stability of ribosomes, boost translation efficiency, and ensure accuracy in protein synthesis [16,84,85]. Moreover, they enable adaptation to environmental stress conditions and potentially contribute to the fine-tuning of ribosomal functions [15]. This evidence suggests that cancer cells may require the accelerated biogenesis of both snoRNAs and snoRNA-associated proteins to sustain cell growth, likely to stimulate small nucleolar ribonucleoprotein (snoRNP)-dependent rRNA modification, accounting for a shift in translational profiles toward the synthesis of growth-promoting and pro-oncogenic proteins.

One intriguing possibility is that specialized ribosomes might contribute to the “hallmarks of cancer”, a set of characteristics that define cancer cells. By selectively translating mRNAs that encode proteins involved in these processes, specialized ribosomes could directly contribute to the malignant phenotype.

For example, pseudouridylation (Ψ) in rRNA involves Watson–Crick base-pairing interactions with the snoRNA H/ACA box that recognize specific uridine residues coupled with dyskerin, which convert uridine into pseudouridine residues. Reduced levels of dyskerin expression in cancer cells are associated with the decreased hypo-pseudouridylation of rRNAs, thereby affecting the structure and function of the ribosome, which is unable to directly translate a subset of IRES-containing mRNAs such as the p27 tumor suppressor, *XIAP*, *TP53*, *CDKN1B*, and the antiapoptotic factor *BCL2L1*, which are all crucial in cell cycle regulation, tumor suppression, and apoptosis [30,71,75].

Barozzi et al. found, in HER2+ and triple-negative breast cancer patients, a negative correlation between many Ψ positions, especially in 28S rRNA, and patient age. Furthermore, certain pseudouridylation sites were significantly associated with histotypes; lymph node metastasis at diagnosis; vascular infiltration by tumor cells; and other biological features like p53 accumulation and BCL-2 and EGFR expression [13].

While it is widely acknowledged that tumor cells tend to exhibit elevated levels of pro-oncogenic rRNA species, the underlying mechanisms and implications of this phenomenon remain subjects of extensive research and investigation [40,42,69,76,77].

Also, 2′-*O*-methylation stands as a predominant post-transcriptional modification in rRNA, significantly impacting ribosome functionality [86]. This modification enriches the complexity of rRNA, thereby playing a pivotal role in ribosome assembly, stability, and protein synthesis [87,88]. This regulation is particularly crucial in cancers where gene expression profiles drastically change [89]. For example, in one study, ribosome profiling of HeLaS3 cells with a loss of methylation in 18S:C174 displayed 1168 transcripts with decreased translation, affecting the cell cycle and mitosis, and 903 transcripts with increased translation related to metabolism, oxidation–reduction, and intracellular transport [74].

The level of 2′-*O*-methylation is associated with different breast cancer subtypes and tumor grades [90]. Also, in aggressive breast cancer cell lines, the suppression of p53 expression leads to changes in the rRNA methylation pattern, impairing translational fidelity and enhancing the translation of key cancer-related genes, namely, *IGF-1R*, *C-MYC*, *VEGF-A*, and *FGF1/2*. This also impacts the differentiation of hematopoietic stem cells, thereby increasing cancer susceptibility [70,71].

Although the precise roles of post-transcriptional modifications on rRNA molecules are not completely understood, it has been proposed that rRNA modifications might increase rRNA half-life and regulate ribosome translation capacity [88], mechanisms that could both positively affect cancer development [71].

The importance of rRNAs in the regulation of gene expression in cancer is also becoming apparent. Biochemical and genetic analyses have shown that changes resulting in either the base-specific or general hypomodification of rRNAs alter translation fidelity, decreasing translation accuracy. rRNA hypomodification is linked to numerous diseases, such as X-DC (X-linked dyskeratosis congenita), several cancers, and aging [23,91,92]. Recently, Babaian et al. identified a cancer-specific single-nucleotide variation in 18S rRNA at nucleotide 1248.U in up to 45.9% of patients with colorectal carcinoma (CRC), and it is present across >22 cancer types [69]. It is thought that these single-nucleotide changes provide an additional level of fine-tuned ribosome structures. Interestingly, quantitative mass spectroscopic analyses have revealed sub-stoichiometric rRNA base modification in normal populations of ribosomes [93,94]. In sum, considering all the mentioned alterations, cancer can be considered an acquired ribosomopathy where structural and functional alterations to ribosomes can sustain neoplastic phenotypes [38].

## 4. Ribosome Diversity as an Entry Point for Targeted Therapy

The idea that ribosomes can be specialized according to various factors, such as cell type, tissue specificity, and physiological stimuli, also suggests that they could be targeted for therapeutic interventions. For example, specialized ribosomes could be selectively targeted without affecting the general population of ribosomes in cells. Also, the control of translation of specific proteins overexpressed or underexpressed in certain diseases, such as cancer or neurodegenerative disorders, could offer an advantage. If ribosome specialization is tissue-specific, therapies could be developed to target ribosomes in specific tissues, thereby reducing systemic side effects. Numerous drugs are known to target the normal ribosome, often resulting in its inhibition. Consequently, it is probable that there are molecules capable of selectively targeting oncoribosome, inducing its selective inhibition and the subsequent death of tumor cells. Understanding how ribosomes change in response to physiological stimuli could allow for dynamic therapies that adapt to the patient’s condition in real time. For many years, technical approaches able to characterize ribosomal heterogeneity at the protein and RNA levels in individual (clinical) samples were lacking. More recently, an increasing number of high throughput approaches, such as Ribomethseq, Hydrapsiseq, RBS, and nanopore sequencing, became available. The consequent mapping of ribosome heterogeneity at the individual level could, in principle, pave the way for personalized medicine where treatments are tailored based on an individual’s specific ribosomal profile. The complexity of ribosome specialization and its regulation would require a deeper understanding before it can be effectively targeted for therapy. Given that ribosomes are fundamental to cellular function, any therapeutic strategies targeting these structures must be approached with caution because of the potential for unintended cellular consequences, requiring the careful design and testing of therapeutic agents. Currently, there remains a significant gap in our understanding regarding specific impacts on oncoribosomes subsequent to the administration of these pharmaceutical compounds.

Figure 1 graphically summarizes the concept of ribosome-tailored therapy in cancer. While the concept is promising, much more research is needed to understand the full scope and implications of ribosome heterogeneity for targeted therapy. It is an exciting avenue that could potentially revolutionize how we approach the treatment of various diseases and conditions.

The development of ribosome inhibitors as a therapeutic approach for treating cancer cells is an emerging field. Recent experimental achievements have confirmed that the ribosome represents a relevant entry point for the development of anticancer drugs. This is due to the reliance of cancer cells on protein synthesis for growth and proliferation.

Examining the crystal structure of homoharringtonin (HHT) in its interaction with the ribosome revealed that HHT becomes integrated into the A-site of the large ribosomal subunit, obstructing access to the charged tRNA, thus interfering with the elongation process in free ribosomes [95]. A key short-term effect of HHT is the rapid depletion of proteins with short half-lives. These short-lived proteins are often critical for cell survival and proliferation. Proteins like c-Myc, Mcl-1, and Cyclin D1 are encoded by mRNAs with complex 5′ UTRs and are particularly vulnerable to HHT action. In hematological malignancies, HHT induces apoptosis by causing the loss of such short-lived proteins, including the downregulation of Mcl-1, an antiapoptotic protein. The depletion of Mcl-1 may lead to an increase in free BH3-only proteins, further promoting apoptosis [96]. HHT treatment also reduces levels of c-Myc, which, in turn, lowers the expression of elongation initiation factors, exacerbating the inhibition of protein translation [97,98,99]. HHT became the first approved drug against chronic myelogenous leukemia targeting the translational apparatus [100].

Verrucarin A, also called Muconomycin A, is a Type D macrocyclic mycotoxin sourced from the pathogenic fungus *Myrothecium verrucaria*. It is known to inhibit protein synthesis by targeting peptidyl transferase activity, an essential process in the elongation phase of protein synthesis [101]. Preclinical investigations have indicated that Verrucarin A’s ability to hinder protein synthesis adversely affects breast cancer cell lines MDA-MB-231 and T47D, as well as leukemia cells, by suppressing their growth [102]. This compound is also associated with the activation of caspases, the induction of apoptosis, and the initiation of inflammatory signaling within macrophages [102]. Through high-throughput screening, Verrucarin A has emerged as a selective suppressor of clear cell renal cell carcinoma (CCRCC) cell proliferation, indicating its potential in targeted cancer treatment [103]. Furthermore, Verrucarin A has been demonstrated to impact the expression of signaling molecules by partially inhibiting IL-8 production in PMA-stimulated promyelocytic leukemia (HL-60) cells. This effect is linked to the suppression of NF-κB activation [104]. Beyond its direct inhibition of protein synthesis, Verrucarin A is capable of prompting a ROS-mediated intrinsic apoptosis mechanism and potentiating TRAIL-induced apoptosis through eIF2α/CHOP-dependent DR5 induction in tandem with ROS generation [101]. These observations highlight potential avenues for also incorporating Verrucarin A into cancer therapy regimens. Nonetheless, the precise interactions and safety profile, especially its toxicity, demand rigorous scrutiny to determine its viability as a therapeutic agent.

Structural analyses of the 80S ribosome–cycloheximide complex and anti-proliferative activity against cancer cells provide a proof-of-principle showing that the human cytosolic ribosome is a promising target to consider for innovative anticancer therapies. This approach introduces a framework for leveraging human cytosolic ribosomes to target proliferating cells. Examining drug interactions in the human ribosome is crucial for precise medical applications in human health, paving the way for future studies focused on developing new anticancer drugs with heightened specificity and reduced overall toxicity [105].

## 5. Discussion and Conclusions

Exploring ribosome heterogeneity provides a groundbreaking avenue for understanding and precisely treating cancer. Traditionally viewed as uniform entities solely accountable for protein synthesis, ribosomes now unveil more intricate roles in cellular function and regulation. This complexity holds particular significance in the realm of cancer, where abnormal ribosomal activity can trigger uncontrolled cellular growth and proliferation.

Advancements in high-throughput sequencing and computational biology have propelled our comprehension of ribosome heterogeneity to unprecedented levels. These technologies empower researchers to meticulously map the multifaceted landscape of ribosomal components, encompassing variations in both ribosomal proteins and ribosomal RNAs, elucidating their distinct impacts on translational control. This level of detail has unlocked novel pathways for targeted therapies capable of precisely modulating ribosomal function to impede cancer progression. Additionally, pinpointing distinctive ribosomal features in various cancer types holds promising diagnostic implications. These molecular “fingerprints” may function as early detection markers or prognostic indicators, thereby augmenting the precision of existing diagnostic methodologies.

Nevertheless, this field encounters challenges. The intricate interplay between ribosomal components and cellular signaling pathways necessitates a systems biology approach for a thorough understanding. Additionally, translating these discoveries into clinically viable therapies demands rigorous validation, encompassing preclinical trials and ethical considerations, particularly when acting on such fundamental cellular machinery.

Hence, ribosome heterogeneity emerges as a promising yet intricate frontier in cancer research. Its role as a driver of disease progression and a potential therapeutic target cannot be underestimated. As we continue to unravel the complexities of ribosomal function and its perturbations in cancer, the prospect of more effective, personalized treatments becomes increasingly tangible. Future research should prioritize validating these findings in clinical settings and exploring synergies with existing treatment modalities, bringing us closer to innovative therapeutic approaches to cancer.

## Figures and Tables

**Figure 1 biomedicines-12-00155-f001:**
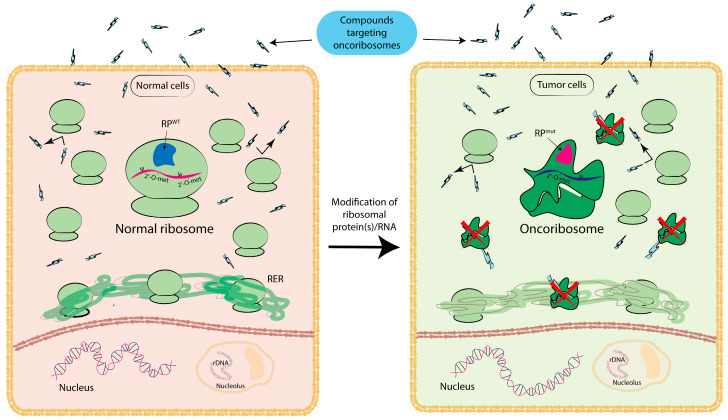
Targeting of ribosomal heterogeneity in normal vs. tumor cells. This figure highlights the potential for selective cancer treatments through ribosome-specific interventions. The left panel depicts normal ribosomal function in normal cells (center, bigger ribosome), where the unaltered protein subunit (RP^WT^) and rRNA of the ribosome are depicted in blue and pink, respectively. In the right panel, altered ribosomes (oncoribosomes) are shown with a mutated protein subunit (RP^mut^), and hypopseudourydinilated and hypomethylated rRNAs are represented in pink and dark blue respectively. In this context, the modification of ribosome components—such as mutations in ribosomal proteins and/or alterations to rRNA post-translational modifications—leads to the establishment of oncoribosome. Compounds specifically designed for oncoribosomes (blue-lightning-shaped icons) can target oncoribosomes, leaving normal ribosomes unaffected. The development of anticancer treatments coupled with ribosome mapping based on an individual’s specific ribosomal profile can pave the way for personalized medicine. Wild-type ribosomal protein, RP^WT^; mutated ribosomal protein, RP^mut^; rough endoplasmic reticulum, RER; ribosomal DNA, rDNA, located in the nucleolus; blue-lightning-shaped icons are an illustrative representation of a specific type of compound or intervention specifically designed to target oncoribosomes.

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
