# Peer review of "Decoding Ribosome Heterogeneity: A New Horizon in Cancer Therapy"

_biomedicines, 2024, doi:10.3390/biomedicines12010155_

Round 1

Reviewer 1 Report

Comments and Suggestions for Authors

This manuscript provides useful knowledge for the cancer research related to ribosome heterogeneity. I do not question the quality of the research presented, but recommend that the manuscript be revised to make much of the information more accessible.

Comments:

1.       In this manuscript, much information is provided on the effects of changes in ribosomal proteins and ribosomal RNA. I propose to tabulate them in an easy-to-understand table.

2.       I suggest that the reference paper be attached to the text on lines 66-68.

3.       What is the "rRNA forms" in line 114? Does this imply a difference in 3D structure of rRNA?

4.       Paragraphs 131-140 do not seem appropriate for this session dealing with normal cells. The next session would be more appropriate.

5.       RLP10 on line 183 is an error for RPL10.

6.       Line 229, X-DC, needs explanation. What does it stand for?

7.       In lines 259-264 and Figure 1, what do you think is the fate of oncoribosome when a specific type of compounds bind to it? Inactivation? Ubiquitination? Degradation? …

8.       In the "References" section, the font type is not consistent. Also, the “doi” of some references are not attached. Please check.

Author Response

Dear Reviewer 1, 

we sincerely appreciate your acknowledgment of the contribution our manuscript makes to cancer research, particularly in the context of ribosome heterogeneity. It is encouraging to hear that the quality of our research is well received. 

In response to your valuable recommendation to enhance the accessibility of the information presented in our manuscript, we are committed to undertaking a thorough revision. Our goal is to ensure that the complex concepts and data are conveyed in a more reader-friendly manner, thereby making our research more approachable to a broader audience, including those who may not specialize in this specific area of cancer research. We welcome any further suggestions you may have and look forward to your guidance in improving our work. 

Thank you once again for your valuable contribution to our research dissemination process. 

Comment 1: In this manuscript, much information is provided on the effects of changes in ribosomal proteins and ribosomal RNA. I propose to tabulate them in an easy-to-understand table. 

Answer: thank you very much for this valuable suggestion, we appreciate your guidance in enhancing the clarity and accessibility of our manuscript. 

In response to your feedback, we agree that presenting this information in a tabular format would significantly improve the manuscript's readability and allow readers to comprehend the complex information more easily. Therefore, we have included a new table in the revised manuscript (page 5, table 1) 

This table is designed to succinctly summarize the key effects of alterations in ribosomal proteins and ribosomal RNA. It is organized to delineate the specific changes and their respective impacts on ribosome function and broader cellular processes. We believe this format will enable readers to quickly grasp the intricate relationships and consequences of these alterations. 

We anticipate that the inclusion of this table will not only aid in better understanding the complex data presented but also serve as a valuable reference that complements the detailed explanations provided in the text. 

Comment 2: I suggest that the reference paper be attached to the text on lines 66-68.  

Answer: Thank you very much. We understand the importance of providing clear and direct references to support our research and arguments. In line with your suggestion, we have now attached the reference paper directly in line 69 in the revised manuscript. This addition will offer readers immediate access to the source material, thereby enhancing the credibility and depth of our discussion in that section. 

Comment 3:  What is the "rRNA forms" in line 114? Does this imply a difference in 3D structure of rRNA?  

Answer: In the context of our manuscript, the term "rRNA forms" refers to the various structural configurations that ribosomal RNA (rRNA) can adopt within the ribosome. This term is indeed intended to imply differences in sequence variation (i.e. 18S rRNA 1248.U line 274 and table 1) and  post-transcriptional modifications of rRNA, which can have significant implications for ribosome function and, consequently, for cellular processes. In line, we changed “rRNA forms” with “rRNA sequence variants” in line 115 in the revised manuscript for more clarity. 

Comment 4: Paragraphs 131-140 do not seem appropriate for this session dealing with normal cells. The next session would be more appropriate. 

Answer: Thank you for your insightful comment. We value your perspective on the organizational flow of the content. 

Upon re-evaluating these paragraphs in the context of your feedback, we agree that their current placement in the section dealing with normal cells may not be the most appropriate. 

Therefore, we moved: 

 “One intriguing possibility is that specialized ribosomes could contribute to the "hallmarks of cancer," a set of characteristics that define cancer cells. By selectively translating mRNAs that encode proteins involved in these processes, specialized ribosomes could directly contribute to the malignant phenotype.” from section “Ribosomes heterogeneity in normal cellular function” to section “The oncoribosome” line 231-234 in the revised manuscript. 

And deleted “Moreover, the tissue-specific expression of specialized ribosomes could have implications for tissue-specific cancers. Recent advancements in ribosome profiling techniques have also opened the possibility of characterizing the translational landscape in cancer cells. Such profiling could identify unique ribosomal translational preferences associated with different cancer types, stages, or treatment responses.” from the text. 

Comment 5:       RLP10 on line 183 is an error for RPL10. 

Answer: Thank you for pointing out the typographical error, we have made the necessary correction in the revised manuscript (line 171). 

Comment 6:       Line 229, X-DC, needs explanation. What does it stand for?  

Answer: We realize that we inadvertently used the abbreviation "X-DC" without providing a proper explanation, which could lead to confusion among our readers. "X-DC" in our manuscript refers to X-linked dyskeratosis congenita and we added the explanation in brackets in line 272 in the revised manuscript 

Comment 7:       In lines 259-264 and Figure 1, what do you think is the fate of oncoribosome when a specific type of compounds bind to it? Inactivation? Ubiquitination? Degradation? …  

Answer: Your question addresses a critical aspect of our research that warrants further clarification.  

We have not explicitly detailed the consequent fate of the oncoribosome upon compound binding as currently, “remains a significant gap in our understanding regarding the specific impacts on oncoribosomes subsequent to the administration of these pharmaceutical compounds”. We added this discussion in the main text line 305-310. 

Comment 8:       In the "References" section, the font type is not consistent. Also, the “doi” of some references are not attached. Please check.  

Answer: Thank you for bringing to our attention the inconsistencies in font type and the omission of some DOIs in the "References" section of our manuscript. We changed the font to be consistent with the rest of the text and added DOI where missing. 

Reviewer 2 Report

Comments and Suggestions for Authors

The manuscript entitles "Decoding Ribosome heterogeneity: A New Horizon in Cancer Therapy." has been submitted as a review by Gelfo et al.

In general, the concept of the review is of great interest. The differently modified ribosomes may represent a crucial link to cancers and could potentially be used as targets for personalized therapy.

However, this possible connection is described in a very general way. There are not enough structural or mechanistic details mentioned in the text of this manuscript to really understand the way the corresponding ribosomes are modified and how this should be the basis for cancerogenesis or putative therapy strategies. Therefore, the authors need to revise the text in order to deliver a clearer presentation of their ideas. 

Moreover, the figure is not very informative. More details could add to the impact of the figure in support of the manuscript text.  

Comments on the Quality of English Language

The extensive editing of English language is required.

Author Response

Dear Reviewer 2, 

Thank you for your insightful feedback on our manuscript. We are pleased to hear that you find the concept of our review to be of interest. 

For these reasons we acted on: 

  1. Expanding on Structural and Mechanistic Details: To address the lack of detailed structural and mechanistic explanations in our manuscript we thoroughly revised the text to include more in-depth information on how ribosomes are modified and the implications of these modifications for cancerogenesis and potential therapeutic strategies. This revision encompassed: 
  • A detailed description of the structural aspects of ribosome modifications. 
  • Elaboration on the mechanistic pathways linking these modifications to cancer development. 
  • A clearer exposition of how these insights could inform personalized cancer therapy approaches. 

Thsi revision involves text changes and new sentences from line 161 to 198, from 206 to 210 from 230 to 246 and from 253 to 262 (highlighted in gray) 

  1. Enhancing the Figure: In response to your feedback on the figure included in our manuscript (Figure 1), we agree that it currently lacks the necessary detail to effectively support our text. Therefore, we acted on the figure design to: 
  • Include more detailed annotations and references to specific parts of the text. 
  • Visually represent the mechanisms and structural aspects of ribosome modifications more clearly. 
  • Ensure that it serves as a useful and informative complement to our revised textual content. 
  1. Improving English Language Quality: We take your comments on the quality of the English language seriously. For these reasons: 
  • We engaged experienced English-speaking colleague for language revision. 
  • We reviewed grammar, syntax, and scientific terminology to enhance clarity and readability. 

Text changes are highlighted in gray 

Round 2

Reviewer 2 Report

Comments and Suggestions for Authors

The authors have adequately addressed my points.